

# MCV-UNet: a modified convolution & transformer hybrid encoder-decoder network with multi-scale information fusion for ultrasound image semantic segmentation

Zihong Xu[1] and Ziyang Wang[2]

[1] Department of Mechanical Engineering, Columbia University, New York, United States of America
[2] Department of Computer Science, University of Oxford, Oxford, United Kingdom

## ABSTRACT

In recent years, the growing importance of accurate semantic segmentation in ultrasound images has led to numerous advances in deep learning-based techniques. In this article, we introduce a novel hybrid network that synergistically combines convolutional neural networks (CNN) and Vision Transformers (ViT) for ultrasound image semantic segmentation. Our primary contribution is the incorporation of multi-scale CNN in both the encoder and decoder stages, enhancing feature learning capabilities across multiple scales. Further, the bottleneck of the network leverages the ViT to capture long-range high-dimension spatial dependencies, a critical factor often overlooked in conventional CNN-based approaches. We conducted extensive experiments using a public benchmark ultrasound nerve segmentation dataset. Our proposed method was benchmarked against 17 existing baseline methods, and the results underscored its superiority, as it outperformed all competing methods including a 4.6% improvement of Dice compared against TransUNet, 13.0% improvement of Dice against Attention UNet, 10.5% improvement of precision compared against UNet. This research offers significant potential for real-world applications in medical imaging, demonstrating the power of blending CNN and ViT in a unified framework.

## INTRODUCTION

Persistent postsurgical pain, defined as pain lasting more than 3–6 months after surgery (*Merskey, 1986*), presents a significant challenge in patient care. Management strategies for postsurgical pain typically encompass symptom control and disease modification (*Kehlet, Jensen & Woolf, 2006*). In clinical practice, the focus often shifts toward symptom control, which primarily relies on the use of narcotics and inhibitors (*Baby & Jereesh, 2017*). The frequent use of narcotics is associated with a range of unwanted side effects, including respiratory depression, nausea, vomiting, and other opioid-related adverse events (*Bajwa & Haldar, 2015*). Moreover, increased narcotic usage has been

Corresponding author
Ziyang Wang,
ziyang.wang17@gmail.com

linked to extended hospital stays and heightened risk of depression (*Armaghani et al., 2016*). Some studies have suggested that indwelling catheters represent an alternative, safe, and effective method for postsurgical pain management (*Wijayasinghe et al., 2016*; *Sola et al., 2012*; *Pacik, Nelson & Werner, 2008*). However, the accurate placement of catheters is crucial, as incorrect placement can lead to unanticipated pain, opioid use, and potential complications, such as readmission or delayed hospital discharge (*Hauritz et al., 2019*). To address these challenges, various methods have been explored to enhance the precision of catheter placement and nerve location identification.

Nerve stimulation (NS) techniques have emerged to enhance the safety and precision of medical procedures, particularly in situations where traditional anatomic landmark techniques may lead to unintended punctures or cannulations (*Pham-Dang et al., 2003*; *Kick et al., 1999*; *Copeland & Laxton, 2001*). NS involves the stimulation of sensory nerves, inducing non-noxious sensations that effectively compete with and attenuate pain signals, thereby reducing pain perception. Additionally, NS has the potential to trigger the release of endorphins, natural pain-relieving chemicals, and modulate nerve activity implicated in pain signaling. Ultrasound technology has gained recognition as an alternative method to elevate the safety and quality of catheter placements in medical practice (*Chan et al., 2003*). Ultrasound techniques offer multifaceted advantages by enabling the visualization of nerve structures before injection, guiding the needle precisely to target nerves, and providing real-time visualization of the local anesthetic's dispersion pattern. Numerous studies have demonstrated the superior performance of ultrasound-guided techniques over traditional anatomic landmarks and NS methods (*Brass et al., 2015*; *Schnabel et al., 2013*). In response to the ongoing pursuit of enhanced nerve identification and catheter placement precision, more advanced techniques have been proposed to further optimize these procedures.

Deep learning-based networks for ultrasound segmentation have emerged as the predominant choice, delivering remarkable segmentation performance at the pixel level. Convolutional neural networks (CNNs) have demonstrated the efficient capacity for extracting intricate features from grid-like data (*Long, Shelhamer & Darrell, 2015*; *Ronneberger, Fischer & Brox, 2015*; *Huang et al., 2020*; *Oktay et al., 2018*; *Li et al., 2022*). The UNet architecture revolutionized CNN-based segmentation with its symmetric encoder–decoder design, enabling impressive results even with limited datasets (*Ronneberger, Fischer & Brox, 2015*; *Wang, Zhang & Voiculescu, 2021*). However, CNNs have inherent limitations due to the localized nature of their convolutional operations, which can lead to under- or over-segmentation in complex ultrasound images. Addressing this challenge, the Attention UNet was introduced, showcasing its efficacy in handling variable small-sized organs by incorporating attention gates (AGs) within the UNet (*Oktay et al., 2018*). Further advancements by researchers like *Chen, Yao & Zhang (2020)* involved the fusion of the ResNet architecture (*He et al., 2016*) with attention mechanisms, thereby enhancing feature extraction and generating high-quality segmentation results for complex features. Another innovative approach, the AAUNet, adaptively selects receptive fields of varying scales from channel and spatial dimensions, leading to substantial improvements in breast lesion segmentation in ultrasound images (*Chen et al., 2022*).

Ultrasound images inherently incorporate non local features, resulting in ambiguous boundaries between target regions and backgrounds. Traditional UNet-based models face challenges in capturing long-range semantic dependencies within ultrasound images (*Fang et al., 2023*). Atrous or dilated convolution methods were introduced in these scenarios (*Chen et al., 2017b*; *Chen et al., 2018*). These methods expand the receptive field of convolutions without increasing the parameter count, allowing them to effectively aggregate multiscale contextual information. Atrous CNN have proven instrumental in achieving more accurate segmentation, particularly in scenarios involving intricate spatial structures and scales (*Yu, Koltun & Funkhouser, 2017*). For instance, *Zhou, He & Jia (2020)* applied atrous convolution to preserve resolution information in feature maps when segmenting brain tumor ultrasound images (*Zhou, He & Jia, 2020*), showcasing the versatility of these techniques in addressing segmentation challenges.

Recent advancements have seen the successful integration of CNN and transformer blocks to preserve global semantic information, with transformers demonstrating exceptional prowess in capturing intricate patterns and relationships in both natural language processing (*Devlin et al., 2018*; *Vaswani et al., 2023*) and computer vision domains (*Parmar et al., 2018*; *Liu et al., 2021*). A novel adaptation of the Transformer for computer vision, known as the ViT, eliminates the need for convolutions to extract features from images (*Dosovitskiy et al., 2020*). The ViT segments images into discrete non-overlapping patches. Spatial positioning information is then introduced to these patches through position encodings, and they are subsequently passed through standard transformer layers. This allows the ViT to effectively model both local and global semantic dependencies. Further augmenting this progress, the Segformer incorporates a Bilinear Fusion mechanism to efficiently merge multi-level feature maps, enhancing both receptive field and resolution for optimized segmentation results (*Xie et al., 2021*). The TransUNet offers a compelling solution with remarkable segmentation performance, effectively marrying high-resolution spatial details from CNN features with the contextual breadth of transformers to address inherent locality limitations and mitigate feature resolution loss, typically associated with pure transformers (*Chen et al., 2021*). The Swin-UNet, by combining a symmetric encoder–decoder structure with skip connections and integrating local-to-global self-attention, marks a significant advancement in image segmentation, optimizing transformer computations and enhancing segmentation efficiency (*Cao et al., 2022*). Lin and collaborators have integrated Swin Transformers and Multi-scale Vision Transformers (*Chen, Fan & Panda, 2021*) into the UNet, fostering excellent long-range dependencies between features of different scales (*Lin et al., 2022*). Additionally, CSwin-PNet was proposed to further enhance long-range dependency modeling, particularly tailored for ultrasound breast segmentation (*Yang & Yang, 2023*).

Given the considerations highlighted, we recognized the significance of global modeling within CNN. Drawing inspiration from TransUNet and atrous convolutions, we introduce our novel approach, referred to as the Modified CNN & ViT hybrid Encoder-Decoder segmentation network with multi-scale information fusion approach (MCV-UNet). To our knowledge, this marks the first endeavor to integrate CNN and ViT explicitly for

ultrasound nerve segmentation. Our contributions in this study can be delineated as follows:

1. Inspired by the burgeoning success of the Vision Transformer in the domain of computer vision, we have further integrated the ViT-layer within the Encoder-Decoder segmentation paradigm, enhancing its feature extraction prowess.

2. Recognizing the importance of capturing intricate details that span from local nuances to broader patterns, we introduce various atrous CNN layers. These layers augment the network's receptive field, bolstering its ability to discern and process multi-scale spatial hierarchies.

3. To validate the efficacy of our approach, we compared MCV-UNet against an array of established baseline methods. The empirical evaluations underscored our network's superior capabilities on a public dataset, yielding competitive results against 15 baseline methods.

The remainder of this article is structured as follows: 'Related work' reviews the relevant literature, highlighting key developments in CNN and ViT utilized to medical image segmentation. 'Approach' details the proposed approach, MCV-UNet, including the network framework, analytical techniques, and related equations. 'Results' discusses the results obtained with MCV-UNet, covering data sources, implementation details, and evaluation criteria. It also provides an in-depth discussion of these results from different perspectives. 'Conclusion' is the conclusion including remarks, summarizing the superior performance of MCV-UNet and suggesting ideas for future research in this field. To aid in the clarity and readability of this article, a table of abbreviations is provided in Table 1.

## RELATED WORK

### Medical image segmentation with CNN

CNN has initially emerged as the predominant methods for image processing tasks (*Milletari, Navab & Ahmadi, 2016*; *Chen et al., 2018*; *Lv et al., 2020*; *Ali, Qureshi & Shah, 2023*). In the domain of medical image processing, where the desired output extends beyond a single class label, the need for precise segmentation of organs or tumors is paramount. Pioneering efforts by *Cireşan, Meier & Schmidhuber (2012)* leveraged deep neural network networks trained on GPUs, leading to substantial improvements in recognition rates on medical image datasets. The introduction of the fully convolutional network (FCN) marked a crucial development by striking a balance between capturing global and local information through the integration of multi-resolution layers (*Long, Shelhamer & Darrell, 2015*). To further enhance training efficiency with limited data, Ronneberger introduced the symmetric UNet architecture, extending the contracting network by incorporating successive layers with skip connections (*Ronneberger, Fischer & Brox, 2015*). UNet quickly gained popularity for its remarkable ability to learn invariance from medical images. LinkNet innovatively directly linked the encoder to the corresponding decoder, ensuring precise predictions without compromising network processing speed (*Chaurasia & Culurciello, 2017*). Subsequent advancements in UNet-based networks, like the Attention UNet with its AGs mechanism, enabled networks to focus on targets

**Table 1  Abbreviation instructions.**

| Abbreviation | Full form |
|---|---|
| CNN | Convolutional neural networks |
| ViT | Vision transformers |
| NS | Nerve stimulation |
| AGs | Attention gates |
| FCN | Fully convolutional network |
| Dice | Dice coefficient |
| Acc | Accuracy |
| Pre | Precision |
| Sen | Sensitivity |
| Spec | Specificity |
| Cost | Computational cost |
| *LN* | Layer normalization |
| *MSA* | Multi-head self-attention |
| *MLP* | Multilayer perceptron |
| *TP* | True positive |
| *FP* | False positive |
| *TN* | True negative |
| *FN* | False negative |

of complex shape and size, expanding its applications in ultrasound image segmentation (*Oktay et al., 2018*). Res-UNet was specifically designed for ultrasound nerve segmentation, enhancing accuracy through the incorporation of dense atrous convolutions and residual multiple posing modules compared to the traditional UNet (*Wang, Shen & Zhou, 2019*). Furthermore, researchers have explored combining recurrent neural networks with residual neural networks to achieve improved organ segmentation performance (*Alom et al., 2018*). Addressing the growing demand for precise medical image segmentation, UNet3+ maximized feature map utilization through full-scale connections (*Huang et al., 2020*). Transfer Learning techniques were also incorporated with the UNet architecture, as demonstrated by *Cheng & Lam (2021)* who applied their network successfully to lung ultrasound segmentation, leveraging mechanisms for detecting edges, shapes, and textures from ultrasound images. Additionally, the Dense-PSP-UNet introduced an innovative Pyramid Scene Parsing (PSP) module, surpassing skip connection settings in performance and employing Contrast Limited Adaptive Histogram Equalization (CLAHE) (*Reza, 2004*) to reduce image noise levels during training (*Ansari et al., 2023*).

## Medical image segmentation with transformers

The transformer architecture, initially pivotal in sequential processing, marked a paradigm shift with its self-attention mechanism, enabling unprecedented performance in various classification tasks (*Vaswani et al., 2017*). In computer vision, the ViT replaced traditional convolutional layers with a novel approach of segmenting images into non-overlapping patches, treated as linear embeddings. This method facilitated contextual relationships between patches through self-attention, enhancing the network's comprehension of the

entire image (*Dosovitskiy et al., 2020*; *Wang, Zhao & Ni, 2022*; *Chen et al., 2021*; *Liu, Hu & Chen, 2023*). ViT has set new benchmarks in object detection (*Fang et al., 2021*), rivaling state-of-the-art CNN architectures, particularly when pre-trained on extensive datasets (*Dosovitskiy et al., 2020*). The introduction of axial (*Ho et al., 2019*) and hierarchical attention (*Yang et al., 2016*) further refined ViT, enabling more precise segmentation. A significant advantage of ViT is its capacity to handle varied image sizes, crucial for intricate ultrasound images. In medical image segmentation, where precision is critical for diagnosis and treatment, traditional methods face challenges like varying contrasts and subtle pathological indicators. TransUNet combined CNN's local detail capture with transformers' holistic view, enhancing the understanding of medical images (*Chen et al., 2021*). Swin-Unet, leveraging transformer blocks, adeptly handles high-resolution medical scans (*Liu et al., 2021*; *Cao et al., 2022*). The hybrid CNN-Transformer network further innovated by integrating large-kernel convolution, effectively capturing multi-scale information (*Liu, Hu & Chen, 2023*). The ViT-Patch introduced a secondary task on the patch tokens, in addition to the primary task on the class token, demonstrating superior performance compared to the standard ViT in breast ultrasound segmentation. token (*Feng et al., 2023*). HA-UNet's introduction of local–global transformer blocks represents a significant step in reducing computational complexity without sacrificing segmentation efficiency (*Zhang et al., 2024*). The inclusion of a cross attention block in HA-UNet not only improved feature integration but also demonstrated significant advancements in ultrasound breast lesion segmentation.

## APPROACH

### Architecture overview

In the domain of deep learning applied to image segmentation, the objective is to map an input image $x$ to its segmented inference $y$. This mapping is denoted as $y_{pred} = f(x; \theta)$, where $f$ is the deep learning network, $\theta$ represents the network's parameters, and $y_{pred}$ is the predicted segmentation of each pixel, where pred $\in [0, 1]$. The corresponding ground truth for the input image $x$ is represented as $y_{gt}$. During the training phase, we use a dataset consisting of batches of paired data represented as $(x, y_{gt}) \in D_{train}$. Our primary aim during training is to optimize the parameters $\theta$ to minimize the difference between $y_{pred}$ and $y_{gt}$. For evaluation on unseen data, we use $(x, y_{gt}) \in D_{test}$ and assess the network's performance by comparing $y_{pred}$ to $y_{gt}$. The MCV-UNet, a novel approach in medical image segmentation for ultrasound images, is depicted in Fig. 1. The architecture, which integrates CNN and ViT, consists of an encoder, bottleneck, decoder, and skip connections. Built upon the foundational UNet structure (*Ronneberger, Fischer & Brox, 2015*), MCV-UNet innovates with key components atrous convolutional and ViT layers. The process begins with two $3 \times 3$ atrous convolutional layers in the encoder, designed to extract multi-scale features while expanding the network's receptive field without significantly increasing computations (*Chen et al., 2017a*). This is followed by standard convolution-based encoders and max-pooling layers, effectively balancing spatial dimension reduction and computational efficiency. In the symmetric design, the feature maps from the encoder aid each upsampling

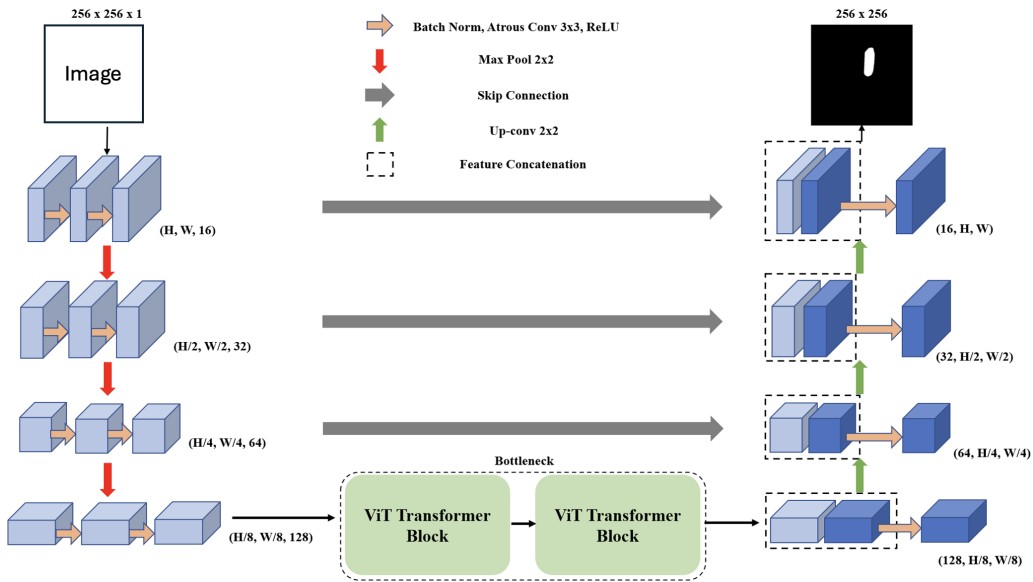

**Figure 1** The proposed encoder-decoder segmentation MCV-UNet based on atrous CNN and ViT blocks.

step in the decoder. A $2 \times 2$ deconvolution layer halves the feature channels, and these are merged with corresponding encoder feature maps *via* skip connections, preserving crucial information.

We introduce a bottleneck with two transformer blocks, leveraging ViT for its segmentation accuracy (*Dosovitskiy et al., 2020*). This unique combination of atrous convolution and ViT allows MCV-UNet to capture both local details and global context effectively, a crucial requirement in medical image segmentation. The final expanding layer in the decoder then maps feature vectors to the desired class numbers, ensuring the output matches the input resolution. MCV-UNet's design is a strategic evolution from conventional UNet, inspired by TransUNet's hybrid CNN-Transformer approach (*Chen et al., 2021*). The specific functionalities of ViTs and atrous convolutions, crucial to MCV-UNet's performance, are further detailed in the subsequent sections.

## Vision transformer layer

Two successive Vision Transformer blocks serves as a key element in the bottleneck between the encoder and decoder is illustrated in Fig. 2. Within each Vision Transformer block, we applied a layer normalization (LN), multi-head self-attention (MSA), a two-layer multilayer perceptron (MLP) with GELU (*Ba, Kiros & Hinton, 2016*; *Hendrycks & Gimpel, 2016*). A residual connection was applied each module (*He et al., 2016*). The computation within these continuous Transformer blocks is illustrated as follows:

$$\hat{z}^l = \text{MSA}\big(\text{LN}(z^{l-1})\big) + z^{l-1} \tag{1}$$

$$z^l = \text{MLP}\big(\text{LN}(\hat{z}^l)\big) + \hat{z}^l \tag{2}$$

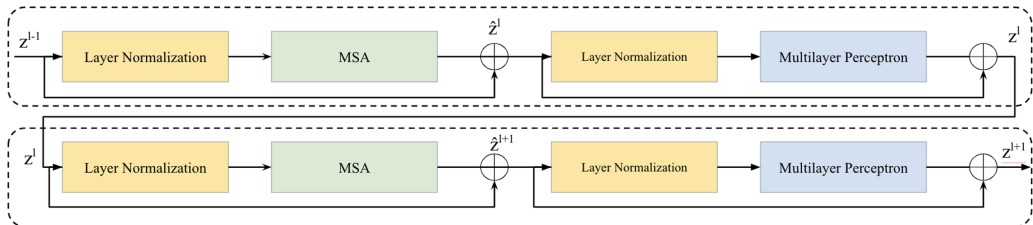

**Figure 2** Two successive Vision Transformer blocks.

$$\hat{z}^{l+1} = \text{MSA}\left(\text{LN}(z^l)\right) + z^l \tag{3}$$

$$z^{l+1} = \text{MLP}\left(\text{LN}(\hat{z}^{l+1})\right) + \hat{z}^{l+1} \tag{4}$$

here, $\hat{z}^l$ and $z^l$ represent the outputs of the MSA module and the MLP module of the $l$th block, respectively.

Drawing inspiration from previous works (*Pan et al., 2022*; *Keles, Wijewardena & Hegde, 2023*; *Wang & Ma, 2023*; *Qin et al., 2022*; *Zhang et al., 2023*), our self-attention computation strategy adheres to the principles of scaled dot-product attention (*Vaswani et al., 2023*). This approach leverages the efficiency of dot-product attention, optimizing the use of matrix multiplication (*Bahdanau, Cho & Bengio, 2014*). The self-attention computation can be illustrated as:

$$\text{Attention}(Q, K, V) = \text{Softmax}\left(\frac{QK^T}{\sqrt{d}} + B\right)V \tag{5}$$

where $Q$, $K$, and $V$ are matrices representing queries, keys, and values, respectively, with dimensions $Q, K, V \in \mathbb{R}^{M^2 \times d}$, where $M^2$ denotes the number of patches in a window and $d$ represents the query or key dimension. The bias term $B$ is derived from the bias matrix $\hat{B}$, with $\hat{B} \in \mathbb{R}^{(2M-1) \times (2M+1)}$.

## Atrous convolution layer

In the classical encoder–decoder architecture, the repeated operations of max-pooling and striding at consecutive layers often lead to a substantial reduction in the spatial resolution of the resulting feature maps. While skip connections and deconvolutional layers can help recover some of this lost spatial information, MCV-UNet takes a further step to mitigate this issue by incorporating atrous convolution within the encoder–decoder architecture.

Atrous convolution, initially introduced for the computation of the undecorated wavelet transform in the "algorithme à trous" scheme (*Holschneider et al., 1990*), has demonstrated high performance in various applications, including semantic segmentation (*Chen et al., 2017a*). In the context of two-dimensional data, atrous convolution is defined as follows:

$$y[i] = \sum_{k}^{K} x[i + rk]w[k] \tag{6}$$

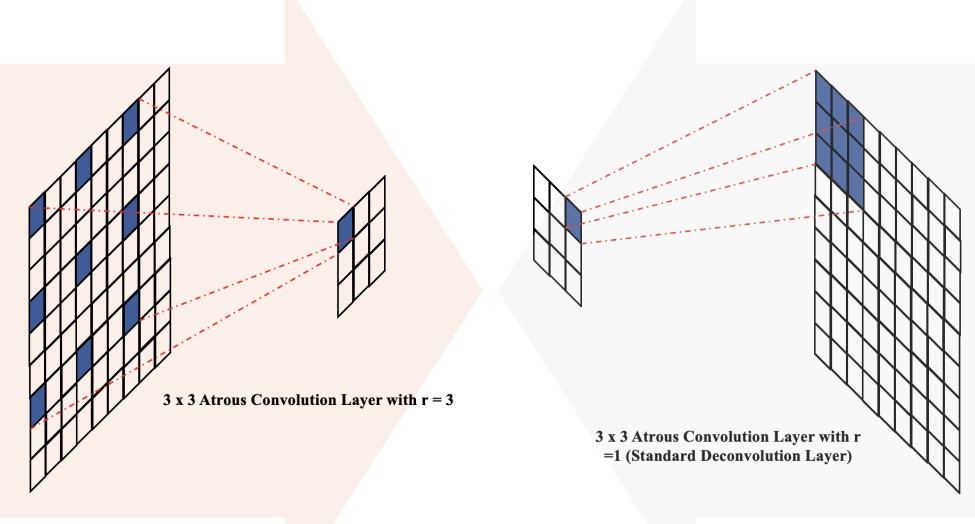

**Figure 3** **The illustration of atrous convolution-based block.**

here, the rate parameter $r$ corresponds to the stride of the sampled input signal, $x[i]$ represents the two-dimensional input signal, and the atrous rate $r$ convolves the input signal with the upsampled filter $w[k]$ by introducing $r - 1$ zeros between consecutive filter values along each spatial dimension (*Chen et al., 2017b*; *Gu et al., 2019*). The parameter $K$ denotes the length of the filter $w[k]$, and the standard convolution corresponds to the special case of an atrous rate $r = 1$. Adjusting the atrous rate provides the network with flexibility in terms of the field of view and enables the generation of larger outputs without significantly increasing computational demands (*Chen et al., 2017a*). Previous works have incorporated atrous convolution in various ways, including within encoder–decoder blocks (*Chen et al., 2018*; *Chen et al., 2017b*), skip connections (*Wang & Voiculescu, 2021*), and the module bridging the encoding and decoding stages to extract dense features (*Lv et al., 2020*; *Gu et al., 2019*; *Pan et al., 2019*; *Ma, Gu & Wang, 2024*).

In the architecture of MCV-UNet, inspired by the encoder–decoder structure with atrous convolution (*Chen et al., 2018*), we placed batch normalization layers before atrous convolution operations with atrous rates $r = 3$ in the encoder blocks and $r = 1$ (standard convolution) in the decoder blocks, as illustrated in Fig. 3. The introduction of holes (with $r = 3$) in the down-sampling process facilitates the computation of responses at all image positions while introducing zeros between filter values. This increases the size of the filter compared to the standard convolution layer, but computations only consider the values of non-zero filter elements, ensuring a constant number of filter parameters and computational operations. Overall, this approach offers the advantage of controlling

**Table 2   The hyper-parameter setting for MCV-UNet and all baseline methods.**

| Epoch | Optimizer | Learning rate | Batch size | Dataset |
|---|---|---|---|---|
| 50 | Adam | $10^{-4}$ | 8 | $5640 \times 256 \times 256$ |

feature resolution, enhancing the receptive field of the network without sacrificing image resolution.

# RESULTS

## Dataset

In our experiments, we employed the Nerve Segmentation database, a publicly available resource from the Kaggle Competition platform (*Anna Montoya et al., 2016*). This database is integral to medical imaging, particularly for the analysis of the brachial plexus nerve, a critical area often studied in ultrasound imaging. This dataset comprises 5,640 $256 \times 256$ ultrasound images, distinctly split into 1,128 testing and 4,512 validation samples, with no overlap between training, validation, and testing sets. Each image encompasses a 2D ultrasound scan of the nerve alongside a meticulously manually annotated 2D segmentation mask, serving as ground truth. The images present a unique challenge due to the random distribution of the nerve within them, demanding precise segmentation skills. To facilitate uniform analysis, all images underwent normalization, scaling pixel values to the range [0, 1], thereby simplifying the task of differentiating the nerve from surrounding tissues. This approach ensures accurate segmentation by leveraging expert annotations and standardized image processing techniques.

## Implementation details

The implementation of our approach was developed using Python 3 and TensorFlow (*Abadi et al., 2015*). Our experiments were conducted on a robust computing setup, featuring an Intel Xeon CPU with 2 vCPUs and 13 GB of RAM, and significantly accelerated with an NVIDIA A100 GPU, equipped with 40 GB of VRAM, known for its high computational efficiency in deep learning tasks. We adapted several networks from established sources, specifically segmentation models and Keras-UNet-Collections, applying necessary modifications to optimize them for our specific dataset. These adaptations were crucial in handling our dataset's unique characteristics.

During the training phase, consistency in parameters across all networks was maintained to ensure fair comparative analysis. We employed the Adam Optimizer (*Kingma & Ba, 2014*), renowned for its efficiency in computing gradients, setting the learning rate to to $10^{-4}$, batch size to 8, and the number of training epochs to 50. Batch normalization layers (*Ioffe & Szegedy, 2015*) were strategically incorporated to enhance training speed and stability. The details of the hyper-parameter setting in the experiment is illustrated in Table 2.

The network's performance was evaluated using the Dice coefficient-based loss, a standard metric in image segmentation tasks, which quantifies the similarity between predicted and ground truth segmentation. We saved the network from the epoch showing

the best performance for testing phase segmentation. On our specified hardware, training a single network typically required 3 to 5 h, depending on the network's complexity and architecture.

## Metrics

To comprehensively evaluate the performance of MCV-UNet, a diverse set of evaluation metrics are utilized. These metrics encompass a range of criteria, including the Dice coefficient (Dice), Accuracy (Acc), Precision (Pre), Sensitivity (Sen), Specificity (Spec), and the parameters of network as computational cost metrics (Cost). Each of these metrics offers a unique perspective on the effectiveness of MCV-UNet in segmenting medical images. The details of our evaluation metrics can be outlined as follows:

$$Dice = \frac{2 \times TP}{2 \times TP + FP + FN} \tag{7}$$

$$Accuracy = \frac{TP + TN}{TP + TN + FP + FN} \tag{8}$$

$$Precision = \frac{TP}{TP + FP} \tag{9}$$

$$Sensitivity = \frac{TP}{TP + FN} \tag{10}$$

$$Specificity = \frac{TN}{TN + FP} \tag{11}$$

where, $TP$ represents the number of true positives, $TN$ denotes the number of true negatives, $FP$ signifies the number of false positives, and $FN$ stands for the number of false negatives.

By employing these diverse metrics, a comprehensive assessment of MCV-UNet's segmentation performance could be achieved. Each metric contributes valuable insights into different aspects of the network's performance, enabling us to evaluate the effectiveness and accuracy of MCV-UNet in the context of medical image segmentation.

## Comparison with state-of-the-arts

To evaluate the performance of MCV-UNet in the context of medical image segmentation, we conducted an extensive comparison with 17 baseline networks, including a diverse range of architectural designs, each with its own strengths and characteristics. To demonstrate the effectiveness of MCV-UNet, we use ViT block bridge the encoder and decoder, with comparisons made against classical CNN networks and their variants. Furthermore, we evaluated the impact of the encoder and decoder design, incorporating atrous convolution layers, by contrasting the result with existing hybrid CNN-ViT networks. The networks compared include: UNet (*Ronneberger, Fischer & Brox, 2015*), UNet-ResNet34

(*Ronneberger, Fischer & Brox, 2015*), UNet-MobileNet (*Ronneberger, Fischer & Brox, 2015*), UNet-InceptionV3 (*Ronneberger, Fischer & Brox, 2015*), Linknet (*Chaurasia & Culurciello, 2017*), Linknet-MobileNet (*Chaurasia & Culurciello, 2017*), FPN-ResNet34 (*Lin et al., 2017*), FPN-MobileNet (*Lin et al., 2017*), TransUNet (*Chen et al., 2021*), FPN-InceptionV3 (*Lin et al., 2017*), VNet (*Milletari, Navab & Ahmadi, 2016*), AttentionUNet (*Oktay et al., 2018*), UNet3+ (*Huang et al., 2020*), U2-Net (*Qin et al., 2020*), RARUNet (*Wang, Zhang & Voiculescu, 2021*), QAPNet (*Wang & Voiculescu, 2021*), and R2UNet (*Alom et al., 2018*). This comprehensive comparison allows us to demonstrate the unique strengths and capabilities of MCV-UNet in the context of medical image segmentation.

## Qualitative results

Figure 4 displays qualitative comparison results with three randomly chosen ultrasound medical images alongside their corresponding ground truths. The original raw images are removed due to Kaggle Policy. For each example, predictions generated by 17 baseline methods are compared with the results from MCV-UNet. The results of these visual analyses yield valuable insights into the performance of each approach. Classical CNN-based methods, including UNet (*Ronneberger, Fischer & Brox, 2015*), Attention UNet (*Oktay et al., 2018*), and V-Net (*Milletari, Navab & Ahmadi, 2016*), tend to exhibit issues of over-segmentation or under-segmentation. For instance, in the first example, UNet-MobileNet over-segments the nerve while V-Net under-segments it. This observation underscores the superior capability of MCV-UNet in effectively encoding global contexts and distinguishing the semantics. In addition, in the context of existing hybrid CNN-ViT networks, the predictions generated by TransUNet (*Chen et al., 2021*) demonstrate coarser characteristics than those by MCV-UNet, particularly with regard to boundary and shape. In the third example, MCV-UNet displays excellent alignment with nerve boundary of the ground truth, whereas TransUNet predicts more false positives. These visual comparisons proves the superior performance of our network, characterized by its capacity to preserve detailed shape information, resulting in fewer false positives and false negatives compared to the baseline methods. This superiority is attributed to the successful combination of CNN and ViT architectures in preserving high-level global information and low-level details, while minimizing spatial information loss with atrous convolution layers.

## Quantitative results

Table 3 presents a comprehensive quantitative evaluation of our MCV-UNet in comparison to the 17 baseline methods in Tables 3, 4, 5. The quantitative results proved the exceptional performance of MCV-UNet on most evaluation metrics. For the main criterion metric of dice coefficient, MCV-UNet achieves a remarkable result of 62.51%, surpassing the second-ranked network by 0.47%. In terms of accuracy and precision, MCV-UNet outperforms all competitors, with a 0.18% increase in Acc and a 1.16% increase in Pre compared to the second-best network. MCV-UNet exhibits competitive performance in sensitivity and specificity metrics, aligning with the top-performing networks. Regarding computational cost, MCV-UNet falls within the median range of all trained networks, showing a slight advantage over our derived architecture, TransUNet. These quantitative results indicate

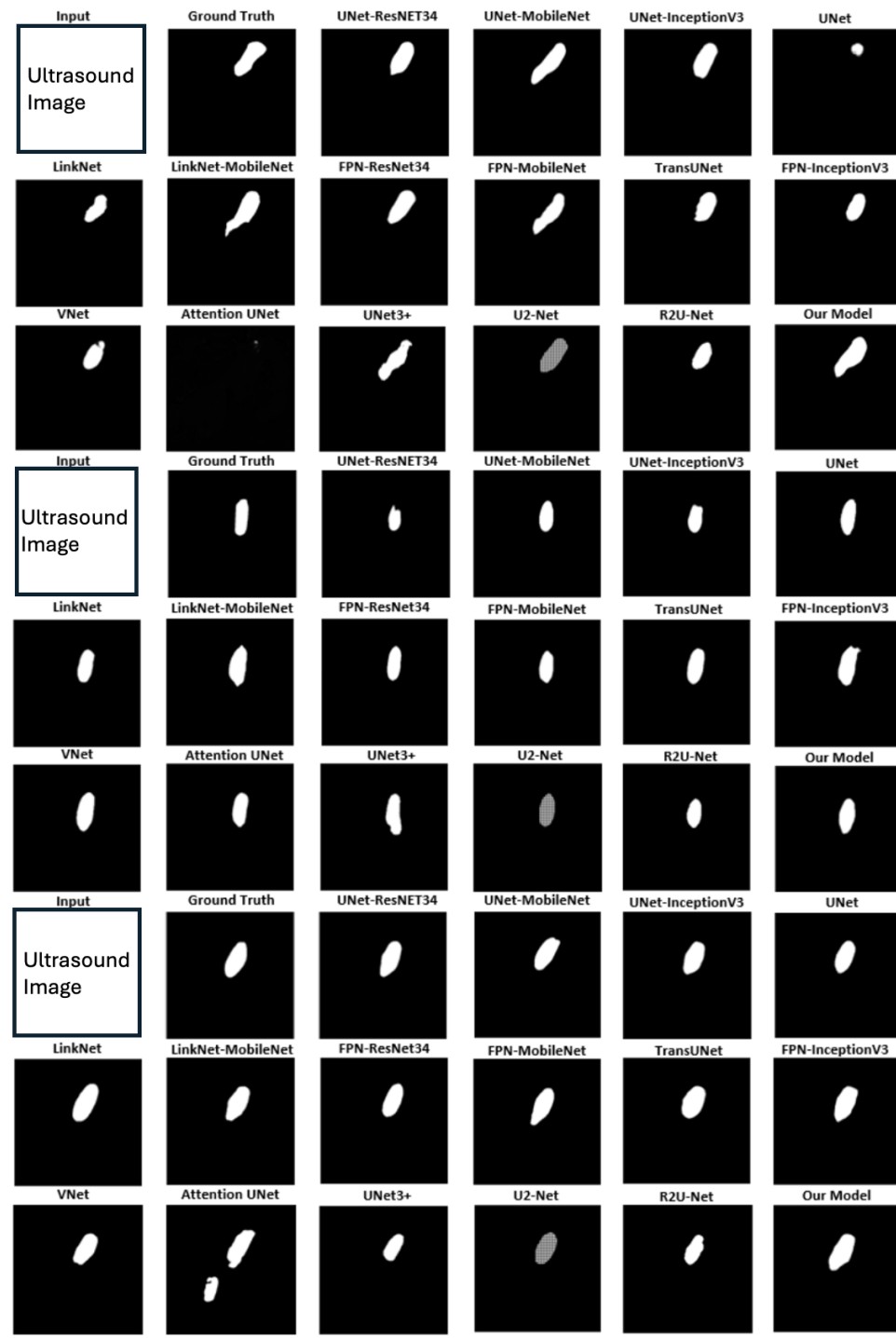

**Figure 4** The segmentation results of different networks on the brachial plexus nerve testing dataset.

**Table 3 The performance of MCVUNet with other baseline methods on ultrasound nerve segmentation test set.**

| Network | Dice | Acc | Pre | Sen | Spec | Cost |
|---|---|---|---|---|---|---|
| UNet | 0.6217 | 0.9910 | 0.6179 | 0.6256 | 0.9953 | 1,967,041 |
| UNet-ResNet34 | 0.6165 | 0.9910 | 0.6218 | 0.6130 | 0.9955 | 24,456,160 |
| UNet-MobileNet | 0.6184 | 0.9907 | 0.6038 | 0.6349 | 0.9950 | 8,336,343 |
| UNet-InceptionV3 | 0.6173 | 0.9899 | 0.5599 | 0.6997 | 0.9934 | 29,933,111 |
| LinkNet | 0.6218 | 0.9911 | 0.6286 | 0.6168 | 0.9956 | 20,325,137 |
| LinkNet-MobileNet | 0.6067 | 0.9896 | 0.5474 | 0.6840 | 0.9932 | 4,546,071 |
| FPN-ResNet34 | 0.6172 | 0.9906 | 0.5941 | 0.6435 | 0.9947 | 23,930,960 |
| FPN-MobileNet | 0.5971 | 0.9907 | 0.6099 | 0.5859 | 0.9955 | 6,103,111 |
| TransUNet | 0.5976 | 0.9888 | 0.5193 | **0.7037** | 0.9922 | 4,675,335 |
| FPN-InceptionV3 | 0.6222 | 0.9904 | 0.5803 | 0.6745 | 0.9942 | 25,029,287 |
| VNet | 0.5955 | 0.9896 | 0.5533 | 0.6491 | 0.9937 | 3,690,129 |
| AttentionUNet | 0.5533 | 0.9887 | 0.5184 | 0.5934 | 0.9934 | 638,322 |
| UNet3+ | 0.5890 | 0.9899 | 0.5695 | 0.6099 | 0.9945 | 497,848 |
| U2-Net | 0.4366 | 0.9892 | 0.5738 | 0.3523 | **0.9969** | 3,775,677 |
| R2UNet | 0.6115 | 0.9910 | 0.6230 | 0.6004 | 0.9956 | 1,448,215 |
| RARUNet | 0.6219 | 0.9910 | 0.6181 | 0.6257 | 0.9953 | 11,793,638 |
| QAPNet | 0.6218 | 0.9910 | 0.6180 | 0.6257 | 0.9954 | 9,472,052 |
| Ours | **0.6251** | **0.9928** | **0.6359** | 0.6912 | 0.9960 | 4,675,329 |

**Notes.**
The best performance results are highlighted in bold. The second-best performance of MCV-UNet is highlighed with an underline.

MCV-UNet's competitive edge across multiple evaluation metrics relative to the 17 baseline networks. The parameters of MCV-UNet is 43%, 76&, 49% lower than UNet with mobilenet as network backbone, LinkNet, and QAPNet. Notably, it is also slightly lower than the current advanced ViT-based TransUNet due to the modified atrous CNN is utilized. They also emphasize capabilities of MCV-UNet of combining the strengths of classical CNN and hybrid CNN-ViT networks.

## Sensitivity analysis & ablation study

In addition to our primary experiments, we carried out a sensitivity analysis focused on the hyper-parameter setting related to the dilated rate in our multi-scale CNN. This analysis is essential to determine the optimal configuration for effectively segmenting nerves in ultrasound images. The detailed results of this analysis are presented in Table 4. These findings validate the effectiveness of the dilated rate settings in our proposed MCV-UNet.

To study the individual and collective impact of the multi-scale modules proposed in our network, we conducted an ablation study, the results of which are detailed in Table 5. This study methodically explores the effects of omitting or modifying various components of our network. These findings not only validate the efficacy of each proposed contribution but also highlight their synergistic effect in enhancing the accuracy and robustness of the ultrasound nerve segmentation process.

**Table 4** The sensitivity analysis of atrous CNN setting.

| Dilation rate | 1 | 2 | 3 | 4 | 5 | 6 |
|---|---|---|---|---|---|---|
| Dice | 0.5976 | 0.6105 | 0.6251 | 0.6248 | 0.6296 | 0.6175 |
| Acc | 0.9888 | 0.9894 | **0.9928** | 0.9912 | 0.9907 | 0.9910 |
| Pre | 0.5193 | 0.5423 | **0.6359** | 0.6323 | 0.5955 | 0.6186 |
| Spec | 0.9922 | 0.9929 | **0.9960** | 0.9957 | 0.9946 | 0.9954 |

**Notes.**
The best performance results are highlighted in bold. The second-best performance of MCV-UNet is highlighed with an underline.

**Table 5** The ablation study of MCV-UNet on ultrasound nerve segmentation test set.

| Multi-Scale CNN | Self-Attention | Dice | Acc | Pre | Sen | Spec |
|---|---|---|---|---|---|---|
| | | 0.6217 | 0.9910 | 0.6179 | 0.6256 | 0.9953 |
| ✓ | | 0.6220 | 0.9903 | 0.5780 | 0.6706 | 0.9941 |
| | ✓ | 0.5976 | 0.9888 | 0.5193 | 0.7037 | 0.9922 |
| ✓ | ✓ | **0.6251** | **0.9928** | **0.6359** | **0.6912** | **0.9960** |

**Notes.**
The best performance results are highlighted in bold.

# CONCLUSION

In this study, we studied the combination of modified CNN and ViT to address the intricate challenge of ultrasound nerve segmentation. Recognizing the limitations inherent in conventional CNN-based networks, especially their restricted capacity to exploit long-range semantic dependencies in ultrasound images, we proposed the MCV-UNet. This novel design represents a modified encoder–decoder framework that seamlessly combines the robust capabilities of both CNN and ViT, while it integrates atrous convolution layers to effectively recover lost spatial information. This kind of multi-scale feature information extraction is valuable in ultrasound imaging, because the nerve structure is complex, and the location, size of nerve can be different in each of ultrasound image. Considering other modalities images, such as CT, MRI, and PET, the MCV-UNet is also valuable to be explored especially when the region of interest (ROI) is complex and should be recognized based on both of the local- and global-based features.

The qualitative and quantitative evaluations indicates that the proposed network outperformed 17 classical baseline methods, exhibiting fewer FP and FN—a testament to its robustness and precision. The integration of various dilated CNN layers further amplified its feature extraction capabilities, bridging the gap between local and global contextual understanding in the images.

Future work might consider refining the network architecture, introducing novel attention mechanisms, or expanding the approach to other challenging medical imaging domains. The computational burden should also be further decreased, because the current parameters of network is still high due to the utilization of ViT. Different types of CNN, network pruning, and knowledge distillation can also be studied to enable the efficiency of the network.

### Funding

The authors received no funding for this work.

### Competing Interests

The authors declare there are no competing interests.

### Author Contributions

- Zihong Xu conceived and designed the experiments, performed the experiments, analyzed the data, performed the computation work, prepared figures and/or tables, authored or reviewed drafts of the article, and approved the final draft.
- Ziyang Wang conceived and designed the experiments, performed the experiments, analyzed the data, performed the computation work, authored or reviewed drafts of the article, and approved the final draft.

### Data Availability

The code is available in the Supplemental File.

The raw data is publicly available at Kaggle: https://www.kaggle.com/competitions/ultrasound-nerve-segmentation.

### Supplemental Information

Supplemental information for this article can be found online at http://dx.doi.org/10.7717/peerj-cs.2146#supplemental-information.

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
