# Peer review of "MCV-UNet: a modified convolution & transformer hybrid encoder-decoder network with multi-scale information fusion for ultrasound image semantic segmentation"

_PeerJ Computer Science, doi:10.7717/peerj-cs.2146_

## Round 0.1 · original submission · Major Revisions

As per comments from three reviewers, I suggest a major revision for this paper.

·

Basic reporting

- Please add an organization related to the article at the end of the introduction section.
- Add more discussion about the proposed method in the revised version.
- The notations and abbreviations used in this article should be summarized in a table for better organization of the text.
- The author should discuss the computational burden of the proposed study in terms of cost and complexity.
- The authors should describe the advantages and disadvantages of the model they have created.
- According to Table 1, the authors have stated that they achieved successful performance with their model compared to other models. However, if this success is in terms of time, then there is no problem. However, if the created model performs poorly in terms of runtime compared to other models and only provides a slight improvement against other models, then it indicates a problem with the temporal complexity of the model. Therefore, information about temporal complexity should be provided.
- Authors, please also specify the values of the external parameters and hyperparameters used in the model in a table format.

Experimental design

no comment

Validity of the findings

no comment

Additional comments

no comment

Reviewer 2 ·

Basic reporting

Good writing and organization of the paper.
Overall the paper is written well. The authors have presented good amount of comparison with other transformer-based segmentation techniques.
However, there is an non necessary use of complicated terms and composition. The article uses some unusual English, but simple words could have conveyed the message better. For example, juxtapose >> compare
Results provided support the claims in the paper.

Experimental design

Is the proposed pipeline also effective for other medical imaging modalities?
CNNs and ViTs have been integrated and combined in many previous studies, including those in medical imaging. The main idea of this approach is to embed a vit in the bottleneck part of an encoder-decoder network. This is not the first work that does it. For example, the review in [1] covers studies in brain imaging that embed the transformer module in the bottleneck part. So, the novelty of this work is limited.

A major shortcoming of the study is the limited experiments on a single type of imaging data, namely the ultrasound images for nerve segmentation tasks. So, the effectiveness of the proposed pipeline cannot be generalized.

Since the results are presented for a publicly available dataset, it would make more sense if the leader board score is also shared.


[1] Artificial Intelligence–Based Methods for Integrating Local and Global Features for Brain Cancer Imaging: Scoping Review, JMIR Medical Informatics, 2023 doi: 10.2196/47445

Validity of the findings

Conclusions are well summarized. Despite limited novelty, the results can be insightful.

Additional comments

Al comments are covered already.

Reviewer 3 ·

Basic reporting

The expression of this artical is clear. However, the brevity and professionalism of the presentation needs to be improved.
For example, there are many repetitive statements in section 3.1: lines 174-176 vs. lines 193-194, line 190 vs. line 214, lines 191-192 vs. lines 218-219.
Some of the less professional expressions such as: lines 233 and 302, 'Where' should be 'where'.
Some relevant prior literature is not mentioned, such as H. Feng, B. Yang, et al., Identifying Malignant Breast Ultrasound Images Using ViT-Patch. Appl. Sci. 2023, 13, 3489. https://doi.org/10.3390/app13063489.

Experimental design

no comment

Validity of the findings

The results in Table 3 should be further analyzed and interpreted, as the table shows that the introduction of self-attention alone leads to a performance degradation instead, which seems to be contradictory to the results of Chen et al 2021b (TransUnet).

Additional comments

The main contribution of this paper is only the introduction of a trous convolution based on the TransUnet architecture. However, there are multiple statements (such as lines 13-15, lines 17-19, and lines 106-108) in the paper that would lead one to believe that fusing CNN and ViT (i.e., the TransUnet Architecture) is one of the main contributions of this paper.

---

## Round 0.2 · accepted · Accept

In the opinions of previous reviewers and mine, this revised paper can be accepted.

·

Basic reporting

Thank you to the authors for fulfilling what I wanted and this article has reached an acceptable level in my opinion.

Experimental design

see above

Validity of the findings

see above

Additional comments

see above

Reviewer 2 ·

Basic reporting

The revised draft is well presented.

Experimental design

In the revised draft, the authors have elaborated the proposed architecture.

Validity of the findings

no comment

Additional comments

It is my understanding that the authors have addressed my comments. I would be happy to recommend the paper for acceptance.

Reviewer 3 ·

Basic reporting

All of my concerns have been addressed in the revised manuscript, and I have no further comments.

Experimental design

no comment

Validity of the findings

no comment